# Epigenetic conservation at gene regulatory elements revealed by non-methylated DNA profiling in seven vertebrates

**Hannah K Long[1,2†], David Sims[3†], Andreas Heger[3], Neil P Blackledge[1], Claudia Kutter[4], Megan L Wright[5], Frank Grützner[5], Duncan T Odom[4,6], Roger Patient[2], Chris P Ponting[3\*], Robert J Klose[1\*]**

[1]Department of Biochemistry, University of Oxford, Oxford, United Kingdom; [2]Weatherall Institute of Molecular Medicine, University of Oxford, Oxford, United Kingdom; [3]CGAT, MRC Functional Genomics Unit, Department of Physiology, Anatomy and Genetics, University of Oxford, Oxford, United Kingdom; [4]Cancer Research UK – Cambridge Institute, University of Cambridge, Cambridge, United Kingdom; [5]School of Molecular and Biomedical Science, The Robinson Institute, University of Adelaide, Adelaide, Australia; [6]Wellcome Trust Sanger Institute, Cambridge, United Kingdom

**Abstract** Two-thirds of gene promoters in mammals are associated with regions of non-methylated DNA, called CpG islands (CGIs), which counteract the repressive effects of DNA methylation on chromatin. In cold-blooded vertebrates, computational CGI predictions often reside away from gene promoters, suggesting a major divergence in gene promoter architecture across vertebrates. By experimentally identifying non-methylated DNA in the genomes of seven diverse vertebrates, we instead reveal that non-methylated islands (NMIs) of DNA are a central feature of vertebrate gene promoters. Furthermore, NMIs are present at orthologous genes across vast evolutionary distances, revealing a surprising level of conservation in this epigenetic feature. By profiling NMIs in different tissues and developmental stages we uncover a unifying set of features that are central to the function of NMIs in vertebrates. Together these findings demonstrate an ancient logic for NMI usage at gene promoters and reveal an unprecedented level of epigenetic conservation across vertebrate evolution.

**\*For correspondence:** chris. ponting@dpag.ox.ac.uk (CPP); rob.klose@bioch.ox.ac.uk (RJK)

†These authors contributed equally to this work

**Reviewing editor**: Anne Ferguson-Smith, Cambridge University, United Kingdom

## Introduction

Short contiguous regions of non-methylated DNA are found associated with most human and mouse gene promoters, where they create a transcriptionally permissive chromatin environment (*Blackledge et al., 2010*; *Thomson et al., 2010*; *Blackledge and Klose, 2011*; *Deaton and Bird, 2011*; *Jones, 2012*) that opposes the repressive effects of DNA methylation (*Klose and Bird, 2006*; *Weber and Schubeler, 2007*). In non-methylated regions, CpG dinucleotide frequency is elevated compared to surrounding sequence (*Bird et al., 1985*; *Bird, 1987*). This is due to accelerated methyl-cytosine mutability, which over evolutionary time leads to a reduction in CpG dinucleotide frequency in densely methylated regions of the genome, while CpG frequency is preserved in non-methylated regions (*Coulondre et al., 1978*; *Bird, 1980*). Taking advantage of the methylation-dependent variations in nucleotide frequency observed in mammals, algorithms were developed to predict non-methylated regions of DNA based primarily on elevated local G+C content and CpG dinucleotide frequency (*Gardiner-Garden and Frommer, 1987*; *Takai and Jones, 2002*).

For more than two decades, CpG island (CGI) predictions (and other nucleotide-based analyses; *Saxonov et al., 2006*) have been used as a proxy for non-methylated DNA in vertebrate comparative

**eLife digest** DNA methylation—the addition of a methyl group to cytosine, one of the four bases found in DNA—is a central process in genetics. By preventing genes from being expressed as proteins, DNA methylation is one of a number of epigenetic mechanisms that can determine which proteins are made in different cell types without changing the underlying DNA sequence.

In warm-blooded vertebrates such as mammals most of the genome is methylated, however short regions of non-methylated DNA are known to be associated with gene promoters (regions of DNA that act as binding sites for the enzymes and transcription factors that transcribe the DNA in the gene into RNA). Much of our current understanding of the role of these islands of non-methylated DNA is based on computational predictions rather than experimental data. In cold-blooded vertebrates, for example, computer models often predict that non-methylated islands are not associated with gene promoters, which potentially suggests an evolutionary divergence in the role of methylation amongst vertebrates. However, this idea has not been confirmed by experimental data.

Long et al. have performed experiments to compare the location of non-methylated islands in seven different vertebrate species. In general they find that computational models are not a reliable method for identifying non-methylated islands. Moreover they find that non-methylated islands are a central epigenetic feature of gene promoters in all vertebrates analysed–including three mammals, a bird, a lizard, a frog and a fish—and not just in warm-blooded vertebrates as suggested by computational models. This shows that the epigenetic function of these non-methylated islands has been conserved over more than 450 million years of evolution.

In addition to the non-methylated islands associated with gene promoters, Long et al. identify two other types: intergenic non-methylated islands that are found away from gene promoters and are said to be 'plastic' because the DNA in these islands can acquire methyl groups, and 'broad' non-methylated islands that span many of the genes that are involved in embryonic development.

By showing that the epigenetic role of non-methylated islands has been conserved over time, and identifying three specific types of island, the work of Long et al. marks an important change in our understanding of epigenetics in vertebrates.

genomics, promoter mapping, and epigenetic studies, often despite little or no experimental evidence that CGIs correspond to bona fide regions of non-methylated DNA outside of mammals (*Ioshikhes and Zhang, 2000*; *Hannenhalli and Levy, 2001*; *Bock et al., 2007*; *Han and Zhao, 2008*). In mouse and human roughly 50–65% of transcription starts sites (TSSs) overlap with CGI predictions. Interestingly, CGI predictions in cold-blooded vertebrates often reside away from gene promoters, with only 16% of zebrafish and 17% of frog TSSs overlapping predicted CGIs. This has led to the suggestion that non-methylated DNA is a unique feature of gene promoters in endotherms, potentially representing a major divergence in the usage of this epigenetic system between warm-blooded and cold-blooded vertebrates (*Aïssani and Bernardi, 1991*; *Sharif et al., 2010*).

Here we experimentally identify non-methylated islands (NMIs) of DNA in the genomes of seven diverse vertebrates, encompassing major evolutionary branch points and including both warm and cold-blooded vertebrates. Interestingly we reveal that CGI prediction does not accurately identify islands of non-methylated DNA, particularly in lower vertebrates. Using our new NMI maps we are able to examine for the first time the relationship between these epigenetically specified features and gene regulatory elements. Interestingly, in contrast to expectation based on CGI predictions in some cold-blooded vertebrates, we now reveal that NMIs are a central and conserved feature of vertebrate gene promoters. Together this work uncovers a unifying set of features that are common to NMI systems across vertebrates and details an unexpected level of epigenetic conservation at vertebrate gene promoters.

## Results

### CGIs poorly predict the location of experimentally determined non-methylated islands in vivo

In order to understand whether the prevailing views about non-methylated DNA function and proposed divergence amongst vertebrate species based on CGI prediction are correct, we isolated genomic

DNA from the testes of seven representative vertebrates and carried out non-methylated DNA profiling using biotinylated CxxC affinity purification (Bio-CAP) (*Illingworth et al., 2010*; *Blackledge et al., 2012*) followed by massively parallel sequencing. We specifically focused our analysis on species covering major evolutionary branch points for both warm and cold-blooded vertebrates, including: two eutherian mammals (human—*Homo sapiens* and house mouse—*Mus musculus*), a monotreme (platypus—*Ornithorhynchus anatinus*), a bird (chicken—*Gallus gallus*), a lizard (green anole lizard—*Anolis carolinensis*), a frog (African clawed frog—*Xenopus tropicalis*), and a teleost fish (zebrafish—*Danio rerio*). Using these new experimentally identified non-methylated island (NMI) maps we initially examined the location of non-methylated DNA across vertebrate genomes by visualising syntenic regions shared among all seven species (*Figure 1A* and *Figure 1—figure supplement 1A and B*). In these regions, the location of computationally derived CGIs in relation to orthologous genes differs greatly across the seven species. In contrast, the distribution of our experimentally identified NMIs was strikingly similar across all seven species. This indicates the suggestion based on CGI prediction that NMIs are a unique feature of warm-blooded vertebrate gene promoters is incorrect (*Aïssani and Bernardi, 1991*; *Sharif et al., 2010*). Based on the disparity between CGI prediction and experimentally-profiled non-methylated DNA, we compared more closely the prediction-based CGI maps with NMIs genome-wide (*Figure 1B*). In human, mouse and chicken, NMIs encompassed most CGI predictions, although in human and mouse roughly 50–60% of experimental NMIs lay outside of predicted CGIs in agreement with recent genome-wide bisulfite sequencing studies (*Molaro et al., 2011*; *Stadler et al., 2011*). Interestingly, the overlap between NMIs and predicted CGIs in lower vertebrates was surprisingly poor revealing that CGI maps in most species fail to accurately identify regions of non-methylated DNA (*Figure 1B*).

## Nucleotide properties within NMIs are variable in different vertebrate genomes

To understand why CGI prediction algorithms often fail, we analysed in detail the nucleotide features of NMIs for all species with a focus on the ratio of observed CpG over expected CpG dinucleotides (CpG O/E) and total G+C nucleotide content (GC content). These are the two features commonly used to identify CGIs genome-wide (*Gardiner-Garden and Frommer, 1987*). We hypothesised that the algorithms may struggle when faced with greatly contrasting genome-wide nucleotide compositions characteristic of diverse phyla. As expected, human and mouse NMIs show elevated CpG O/E and GC content compared to control regions of the genome (*Figure 1C*). However, many human and mouse NMIs have lower CpG O/E and GC content than the CGI predictions, explaining why the CGI predictions do not accurately identify all NMIs in these species. CGI predictions in chicken are surprisingly accurate. This appears to be due to the fact that NMIs in this species have the highest CpG O/E and GC content compared to the surrounding genome amongst all the vertebrates examined. Although platypus NMIs have a similarly high CpG O/E and GC content, its genome is on average more CpG and GC rich than chicken. This causes the algorithm to massively over-predict CGIs. Lizard and frog encode NMIs with CpG O/E content similar to those in mammals, but with lower GC content. In zebrafish, NMI CpG O/E is high yet the GC content is almost indistinguishable from surrounding DNA sequence (*Cross et al., 1991*). Again, in these species this leads to a general failure of CGI prediction to accurately identify NMIs.

The failure of CpG island prediction algorithms to accurately identify NMIs in different species is almost certainly dependent on the variation in CpG density and G+C content amongst vertebrate genomes, but also will rely on genome assembly and annotation quality, particularly of repetitive elements. Indeed, based on genome variations in CpG and G+C content, it has been suggested previously that species-specific CpG island annotation may be required (*Glass et al., 2007*). These sequence variations between species are likely driven by the relative strengths of two processes: reductions of G+C content due to imperfect repair of spontaneous 5-methylcytosine deamination events (*Coulondre et al., 1978*; *Bird, 1980*) and increases in G+C content in species and genomic regions that are especially prone to GC-biased gene conversion, an outcome of recombination (*Duret and Galtier, 2009*). Species differences in these antagonistic processes are likely to have caused the varying levels of G+C content both among vertebrates and across different regions of most amniotic genomes. Unlike CGI predictions, Bio-CAP identifies NMIs through an affinity based isolation of non-methylated CpGs and therefore does not solely rely on nucleotide content in the same way prediction algorithms do. Nevertheless, we considered whether the efficiency of NMI identification by Bio-CAP among species differs due to

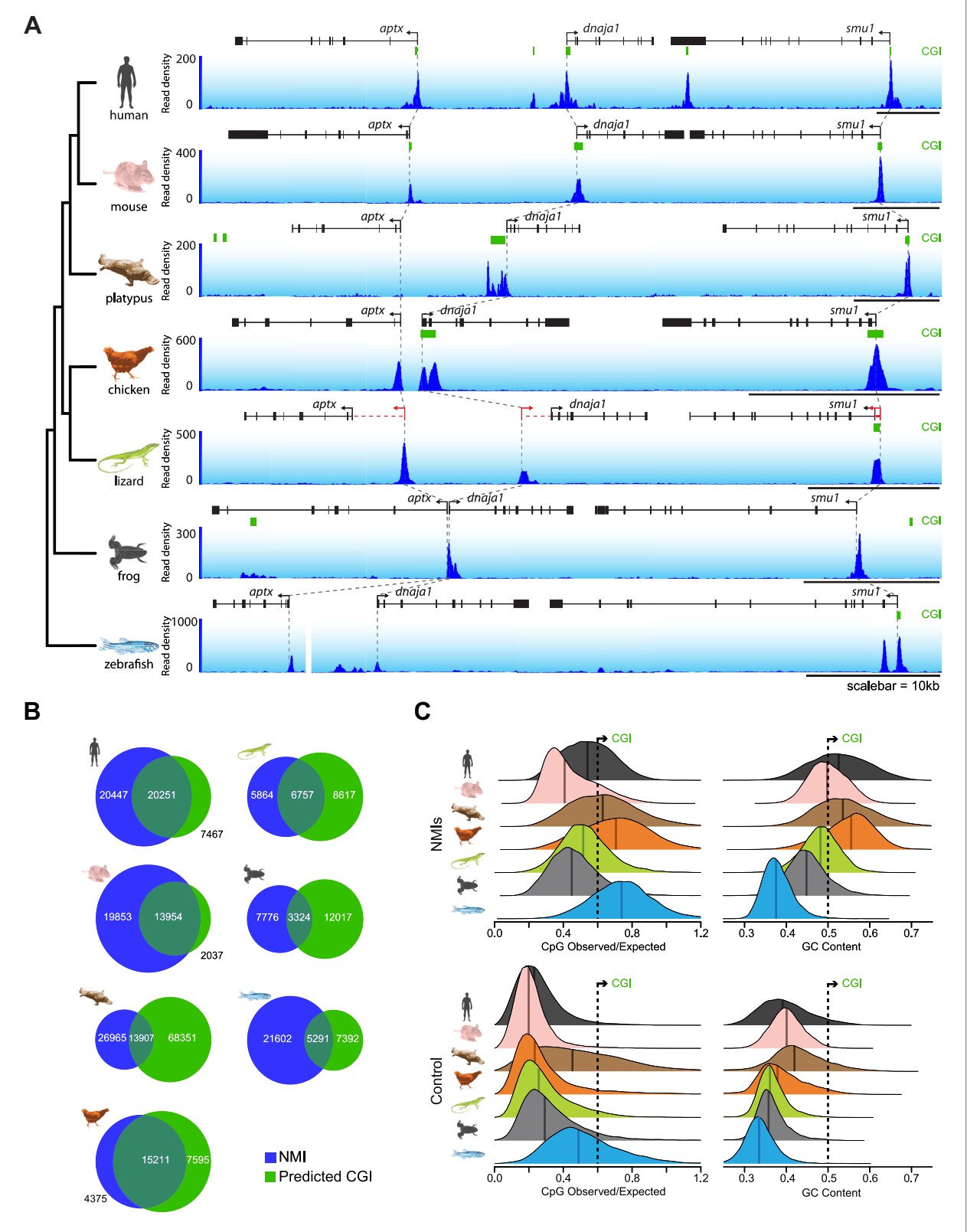

**Figure 1**. CpG island predictions do not accurately identify non-methylated islands of DNA in vertebrate genomes. (**A**) Non-methylated DNA profiles in testes at a representative syntenic region for seven vertebrate species. Genes are shown in black (improved annotation of gene TSSs using RNA-seq data is shown in red), CpG island predictions in green (CGI), and non-methylated DNA profiles are shown in blue. A phylogenetic tree (left)
*Figure 1. Continued on next page*

*Figure 1. Continued*

highlights the evolutionary relationship among the seven species. Dashed grey lines highlight the relationship between the gene TSSs across the species. A gap in the zebrafish profile indicates that *aptx* is found at a separate locus from *dnaja1* and *smu1*. (**B**) The genome-wide overlap between CpG islands (green) and non-methylated islands (blue) is depicted as a Venn diagram for each of the species. (**C**) Nucleotide properties of non-methylated islands and control regions are depicted as density plots. CpG observed/expected (left) and GC content (right) are shown for NMI and control regions of the genome. Median values are shown as dark vertical lines. Thresholds for CpG island prediction are indicated (black dashed line).

The following figure supplements are available for figure 1:

**Figure supplement 1**. NMIs are a conserved feature of vertebrate promoters as illustrated by two syntenic loci.

non-methylated CpG content and density. In contrast, non-methylated DNA fragments, even with low CpG density, are effectively detected by Bio-CAP (*Blackledge et al., 2012*) and a broad distribution of CpG density within NMIs is identified in all species. Therefore, although CGI prediction does function with some degree of accuracy in mammals and bird, CGI prediction maps are in general a poor indicator of where NMIs exist in vivo, presumably due to varying nucleotide content amongst diverse phyla. In light of the fact that CGI prediction maps largely fail to accurately detect experimentally-identified non-methylated regions of DNA, work over the past 25 years that has extensively used these maps as a proxy for non-methylated DNA and evolutionary comparison clearly requires re-evaluation.

## NMIs are a highly conserved feature of vertebrate gene promoters

Using our new genome-wide maps of non-methylated DNA, we first set out to directly examine whether NMIs are a specific feature of warm-blooded vertebrate gene promoters as has previously been suggested (*Aïssani and Bernardi, 1991*; *Sharif et al., 2010*). In human, mouse, and chicken we observed that most TSSs of protein coding genes overlap with an NMI (72%, 66%, 58%, respectively) (*Figure 2A*). In zebrafish, a cold-blooded vertebrate, CGI predictions overlap with only 17% of TSSs suggesting that this is not a major feature of gene promoters in this species. In stark contrast, our new NMI maps reveal strong overlap of NMIs and TSSs (55%) in zebrafish. The occurrence of promoter-associated NMIs in lizard and frog appeared initially to be low. However, by using RNA-seq

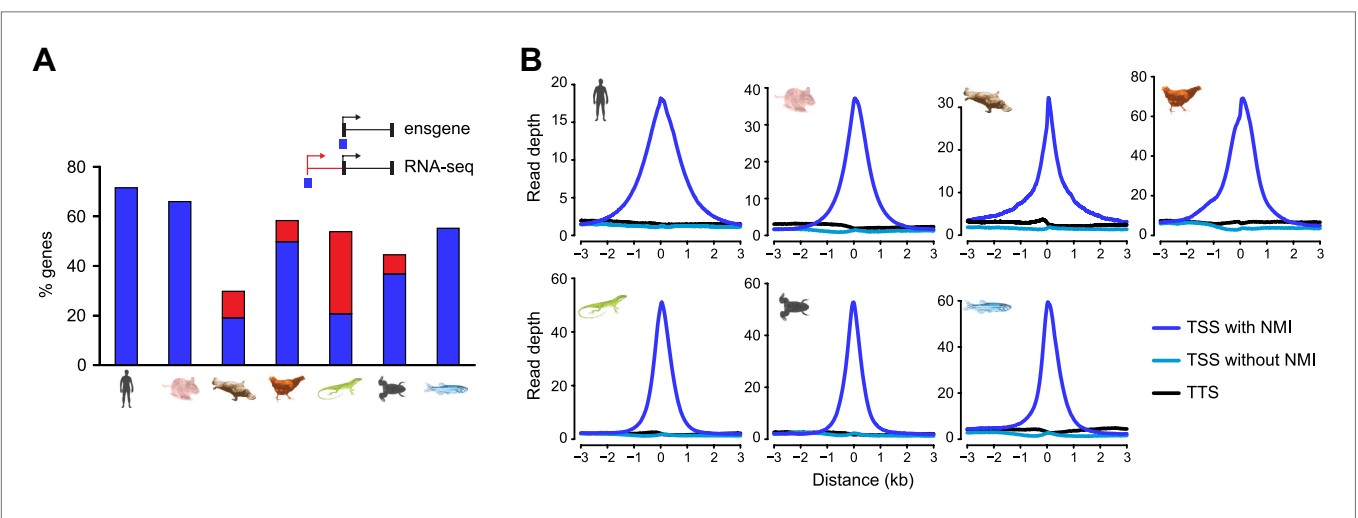

**Figure 2**. Non-methylated islands are associated with gene promoters in vertebrate genomes. (**A**) A histogram depicting the proportion of protein-coding transcription start sites (TSSs) which are overlapped by an NMI for all seven species. Blue bars indicate overlap with annotated TSSs and red bars indicate overlap with additional TSSs identified using RNA-seq data (platypus, chicken and lizard) or Xtev gene sets (frog). (**B**) Profiles of non-methylated DNA were plotted over a 6-kb window centred on all TSSs with an NMI (dark blue), without an NMI (blue), and for all transcription termination sites (TTS, black). The non-methylated DNA signal peaks at the TSS of gene promoters in all vertebrates.

information to refine TSS annotation (*Akkers et al., 2010*; *Barbosa-Morais et al., 2012*), the fraction of TSSs overlapping NMIs in these species increased to 45–58%, similar to that observed in other vertebrates (*Figure 2A*). In platypus the association of NMIs with gene promoters appears lower and does not improve substantially using RNA-seq information (*Brawand et al., 2011*; *Julien et al., 2012*) (19% increases to 30%); however this is likely an inaccurate representation due to a fragmented and incomplete genome assembly which limits our capacity to build accurate gene models. Nevertheless, non-methylated DNA signal centres at TSSs in all seven species (*Figure 2B*). Therefore, in direct contradiction to the contention from previous work (*Aïssani and Bernardi, 1991*; *Sharif et al., 2010*) and CGI prediction maps that non-methylated islands of DNA are a unique feature of warm-blooded vertebrate gene promoters, we now unequivocally demonstrate that NMIs are instead a central and ancient feature of vertebrate gene promoters.

Although DNA sequence within gene regulatory elements is often conserved across vertebrate species, it remains almost completely unknown whether epigenetic features are subject to a similar selective pressure. Interestingly, some evidence has emerged recently indicating that certain epigenetic features may be conserved between mammalian species (*Shibata et al., 2012*; *Xiao et al., 2012*). Taking advantage of our new NMI maps, we investigated whether NMIs are conserved at the TSSs of orthologous genes in the seven vertebrate species analysed. Pairwise comparison of homologous vertebrate gene promoters revealed a surprisingly high propensity for orthologous genes to have an NMI at their respective promoters (*Figure 3A*). This is exemplified by the fact that over 90% of NMIs are shared at orthologous gene TSSs between human and each of the other six species examined. Furthermore, a more extensive three-way comparison between human, mouse and zebrafish revealed conservation of NMIs at nearly all gene promoters of one-to-one orthologues, despite 450 million years of divergent evolution (*Figure 3B*). Therefore, this remarkable degree of conservation reveals a concerted evolutionary drive not only to select for DNA sequence at gene regulatory elements but also to retain NMI identity as an epigenetic feature of gene regulatory elements.

## Intergenic NMIs are associated with distal regulatory elements, non-coding RNAs, and unannotated transcripts

In addition to gene associated NMIs, an unexpectedly large proportion of vertebrate NMIs lie within intergenic regions (*Figure 4A*), suggesting that some NMIs may be functioning in regulatory roles away from known gene promoters. To try and understand how these intergenic NMIs contribute

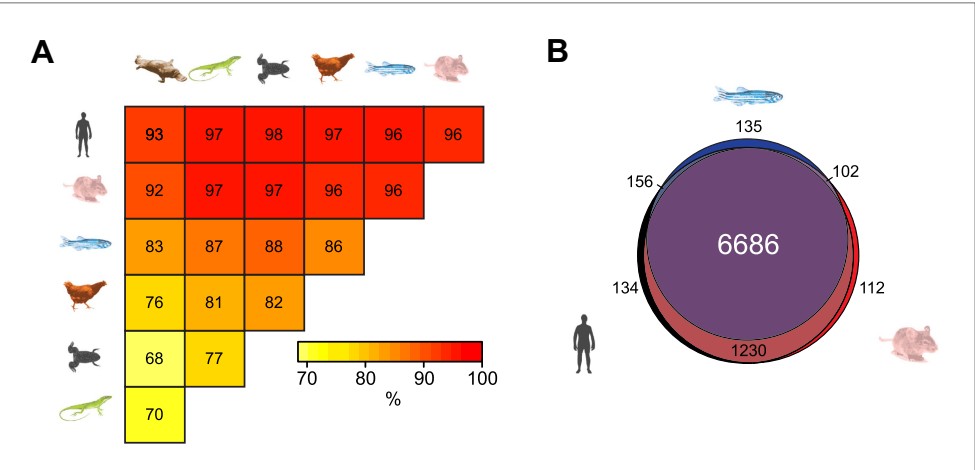

**Figure 3**. Non-methylated islands are a highly conserved epigenetic feature of vertebrate gene promoters. (**A**) The presence of NMIs at orthologous gene TSSs is preserved as illustrated by a pairwise analysis of NMIs at vertebrate gene orthologues. The percentage of NMIs conserved at orthologous gene TSSs was calculated in a pairwise manner and found to be highly statistically significant for all comparisons across the seven vertebrate species ($p < 10^{-10}$, hypergeometric test). (**B**) A proportional Venn diagram illustrating the three-way comparison of NMI presence at conserved human-mouse-zebrafish gene orthologue TSSs.

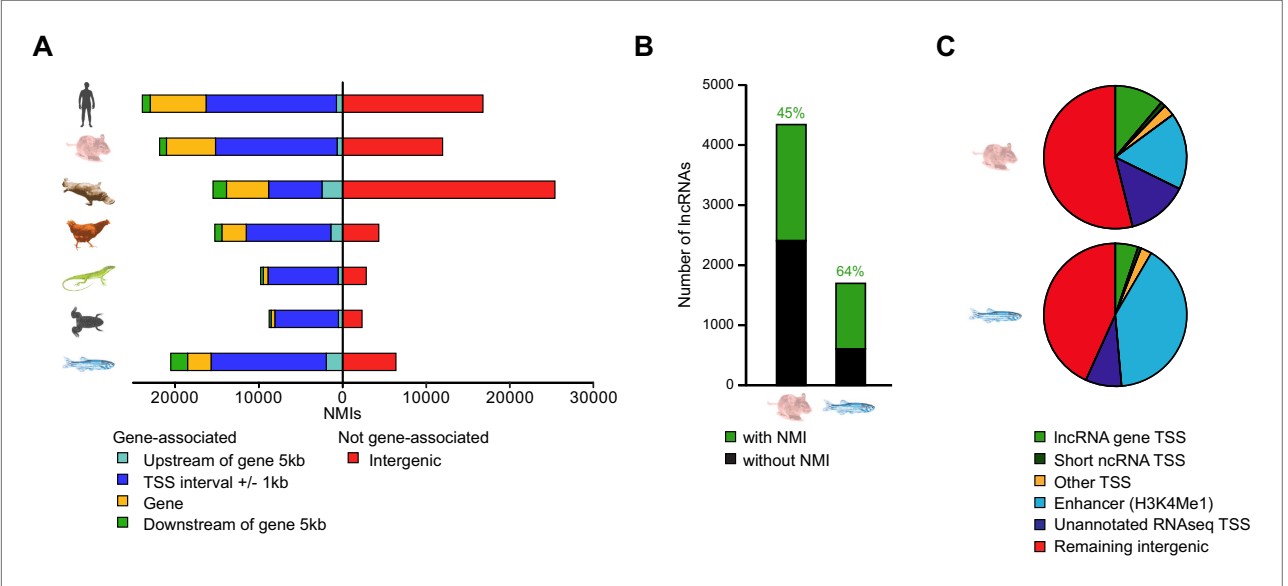

**Figure 4**. Intergenic NMIs are associated with distal regulatory elements, non-coding RNAs, and unannotated transcripts. (**A**) Most NMIs are associated with known protein-coding genes (left) but a substantial proportion are located within intergenic regions of the genome (right). (**B**) NMIs (green) are found at 45% and 64% of all known long non-coding RNA (lncRNA) TSSs (black) in mouse and zebrafish respectively. (**C**) A pie chart depicting the proportion of intergenic NMIs (>5 kb from a protein-coding gene) associated with different genomic features in mouse embryonic stem (ES) cells and zebrafish 24 hpf embryos. The association was performed hierarchically in the following order: lncRNA TSSs, other non-coding RNA TSSs (miRNAs, rRNAs, snRNAs, or snoRNAs), other TSSs (pseudogenes and processed transcripts), putative enhancer mark H3K4me1 and novel RNA-seq TSSs. This analysis indicates that intergenic NMIs mark novel transcriptional units or regulatory elements.

to genome function, we explored whether intergenic NMIs correspond to regulatory elements such as non-protein coding transcripts or enhancers. Recently, histone H3 lysine 4 mono-methylation (H3K4me1) has been found to be strongly associated with enhancer elements (*Barski et al., 2007*; *Heintzman et al., 2007*; *Aday et al., 2011*). In order to allow comparison with both non protein-coding transcript models and available genome-wide H3K4me1 data (*Aday et al., 2011*; *Stadler et al., 2011*) we generated NMI maps for mouse embryonic stem (ES) cells and zebrafish 24 hr post-fertilisation (hpf) embryos. Initially we analysed whether intergenic NMIs were found associated with the TSSs of annotated non-coding RNAs (ncRNAs) and found that less than 3% overlapped with the primary transcripts of miRNAs, rRNAs, snRNAs, or snoRNAs (data not shown). However, in mouse and zebrafish 45–64% of known lncRNA TSSs (*Guttman et al., 2010*; *Belgard et al., 2011*; *Ulitsky et al., 2011*; *Pauli et al., 2012*) were associated with NMIs suggesting that this class of candidate regulatory RNAs are subject to similar epigenetic regulatory processes as protein coding genes (*Figure 4B*). In total, association with the TSSs of all known non-coding transcripts accounted for 15% and 9% of intergenic NMIs in mouse and zebrafish respectively (*Figure 4C*). Intergenic NMIs not associated with annotated non-coding RNAs were then compared with H3K4me1 peaks and a further 17% and 40% of the NMIs, in mouse and zebrafish respectively, corresponded to putative enhancers. Finally, the remaining intergenic NMIs were compared to tissue-specific RNA-seq datasets (*Stadler et al., 2011*) to determine whether they overlapped with un-annotated transcriptional initiation. This revealed that up to 14% and 8% of NMIs in mouse and zebrafish overlap with uncharacterised transcribed regions of the genome. Therefore a substantial proportion (46–57%) of NMIs away from the 5′ end of protein coding genes encompass known ncRNA TSSs, putative enhancers, and other transcribed regions, a property which is highly conserved across two diverse vertebrate species. Importantly, this indicates that NMI profiles provide not just a useful tool for resolving TSSs for protein coding genes in organisms where genome annotation is sparse, but also a valuable resource for identifying putative enhancers or un-annotated coding and non-coding genes. Together these observations reveal that vertebrate NMIs may have an unexpectedly important role away from gene promoters in contributing to the regulatory potential of the genome.

## Differentially methylated islands are found away from gene promoters

In general NMIs in mammals are thought to be maintained in the non-methylated state in most tissues, even if their associated gene is not appreciably transcribed. Recently this belief has been challenged and it appears that some NMIs in human and mouse are more susceptible to differential methylation during tissue specification (*Weber et al., 2007*; *Farthing et al., 2008*; *Mohn et al., 2008*; *Straussman et al., 2009*; *Illingworth et al., 2010*; *Maunakea et al., 2010*). However, it remains unclear whether differential methylation of NMIs is a common strategy for epigenetic regulation in vertebrates. Therefore to address this question we profiled NMIs in liver for all seven species and compared these maps to those already generated for testes. We first separated a core set of NMIs that are shared between the two tissues from a second set of NMIs that are unique to either testes or liver. Unique NMIs from liver or testes were validated by bisulfite sequencing for mouse and zebrafish (*Figure 5—figure supplement 1*). The existence of unique NMIs in each of the interrogated tissues suggests that all vertebrates utilise differential methylation to epigenetically regulate a subset of NMIs. To examine in more detail the properties of shared and unique NMIs, heat maps illustrating non-methylated DNA signal were generated for NMI intervals, ranked by NMI length and clustered according to their classification as shared or unique (*Figure 5A* and *Figure 5—figure supplement 2A*). Interestingly, it was immediately apparent that shared and unique NMIs stratified into two largely distinct classes. Shared NMIs tended to be longer, have higher CpG density (*Figure 5C,D* and *Figure 5—figure supplement 2C,D*) and are generally found associated with the TSSs of protein coding genes (*Figure 5B* and *Figure 5—figure supplement 2B*). This is consistent with the classical view that mammalian NMI-associated TSSs are generally refractory to DNA methylation in most tissues (*Illingworth et al., 2010*). We refer to these as 'canonical' NMIs. In contrast, tissue-specific NMIs that are differentially methylated tended to be shorter, have lower CpG density and are found away from protein-coding gene TSSs (*Figure 5B–D* and *Figure 5—figure supplement 2B–D*). For the subset of differentially methylated TSS-associated NMIs, expression differences of the associated gene were compared between testes and liver for human, mouse, platypus and chicken (*Brawand et al., 2011*). This revealed that genes differentially expressed between liver and testes were significantly enriched for genes with tissue-specific NMIs (*Figure 5* and *Figure 5—figure supplement 3*). Therefore, the presence of DNA methylation closely correlates with gene repression, suggesting that although an infrequent event, methylation-mediated silencing at NMI promoters is a general feature in epigenetic regulation of certain vertebrate protein-coding genes.

## Chromatin modifications at differentially methylated NMIs are dependent on the underlying DNA methylation state

Most tissue-specific NMIs are intergenic, indicating that differential methylation of these elements in vertebrate genomes is generally found away from known gene promoters (*Figure 5B* and *Figure 5—figure supplement 2B*). We have already shown that intergenic NMIs are enriched for regulatory elements (*Figure 4B,C*), therefore one exciting possibility is that differential methylation at NMIs away from gene promoters could function as part of an epigenetically driven regulatory process. Recently, we and others have demonstrated that NMIs are recognized by a class of ZF-CxxC DNA binding proteins that recruit chromatin modifying enzymes to create a unique chromatin environment including placement of H3K4me3 (*Blackledge et al., 2010*; *Thomson et al., 2010*). To examine whether differential methylation of NMIs corresponds to alterations in H3K4me3, the levels of this modification were examined in testes and liver in both human and mouse. This revealed a very clear segregation of H3K4me3 with the tissue that has the unique NMI (*Figure 6A*). Similarly, when we extended this analysis to the lower vertebrates frog and zebrafish there was a clear enrichment of H3K4me3 at tissue-specific NMIs (*Figure 6B*). Together these observations demonstrate that differential DNA methylation at gene promoters is associated with gene repression and that the chromatin modification state of differentially methylated NMIs may be regulated in a switch-like mechanism dependent on the underlying DNA methylation status. This epigenetically 'plastic' subset of NMIs suggests that the landscape of non-methylated DNA in vertebrates may contribute an unexpected level of functional diversity across different tissues in an otherwise static DNA sequence.

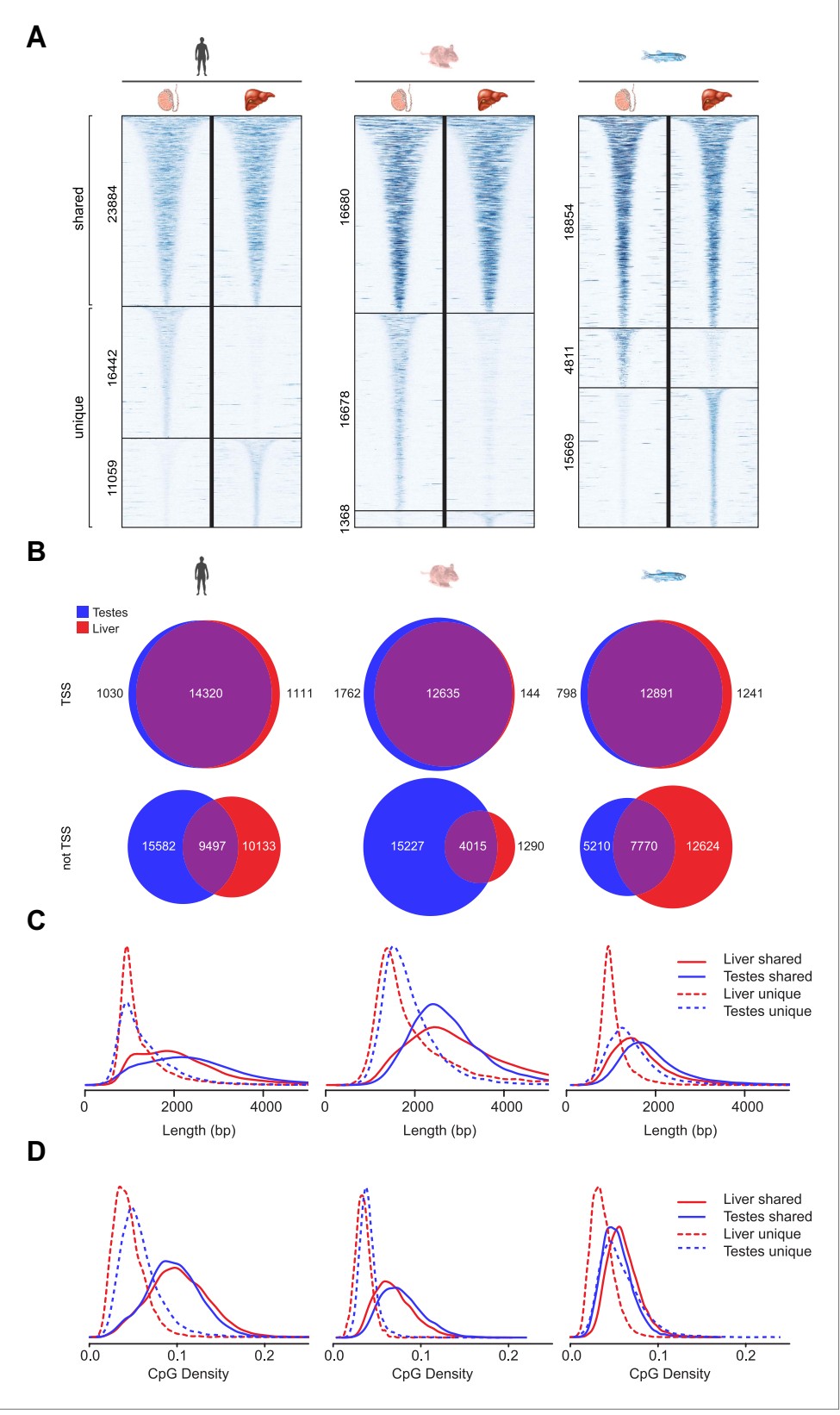

**Figure 5**. Differential methylation of a subset of NMIs. (**A**) All vertebrate genomes have a subset of NMIs that are subject to differential methylation as illustrated by a heat map of non-methylated DNA signal from testes and liver

*Figure 5. Continued on next page*

*Figure 5. Continued*

in human, mouse and zebrafish. In each case NMIs are ranked according to length and clustered as shared (upper) or unique (lower) between the two tissues. A 5-kb window centred at the NMI is shown and read density is indicated by colour intensity. (**B**) The overlap of NMIs identified in liver and testes is depicted by Venn diagrams for NMIs associated with protein-coding TSSs (upper) and for NMIs away from TSSs (lower). NMIs at TSSs are generally non-methylated in both tissues whereas differentially methylated NMIs tend to be found away from TSSs. (**C**) NMI length distribution plots for shared (Shared NMIs, solid line) or unique (Unique NMIs, dashed line) NMIs from testes (blue) or liver (red). Shared NMIs tend to be longer than tissue-specific unique NMIs. (**D**) CpG density distribution plots for shared (solid line) or unique (dashed line) NMIs from testes (blue) or liver (red). Shared NMIs tend to have higher CpG density than unique NMIs.

The following figure supplements are available for figure 5:

**Figure supplement 1**. Validation of differentially methylated NMIs between liver and testes in mouse and zebrafish by bisulfite sequencing.

**Figure supplement 2**. Differential methylation of NMIs in platypus, chicken, lizard and frog and length distributions of NMIs from all seven vertebrates.

**Figure supplement 3**. Genes with TSS-associated testes or liver specific NMIs are over-represented for increased differential expression in the same tissue.

## A unique class of 'broad' NMIs are associated with developmental genes and subject to polycomb regulation

The majority of vertebrate genes have a punctate peak of non-methylated DNA at their TSS in fitting with the canonical view of an NMI (*Figure 2B*). Interestingly however, we observed a subset of genes that deviated from this generality and tended to be encompassed by broad regions of non-methylated DNA ('broad NMIs') that often covered the majority of the gene. This striking observation is exemplified by the *sp9* and *otp* genes (*Figure 7A* and *Figure 1—figure supplement 1A*) and more complex gene clusters including the *hox* loci (*Figure 7—figure supplement 1*). Like canonical NMIs, broad NMIs appear to be H3K4me3 modified over the entire non-methylated region indicating they are also targeted by ZF-CxxC dependent chromatin modifying activities (*Figure 7B,C*). To begin trying to identify common features shared amongst genes associated with broad NMIs, Gene Ontology (GO) analysis was performed (*Figure 7D*). Interestingly, genes encompassed by broad NMIs are highly enriched for transcription factors and genes involved in development, suggesting that this epigenetic feature may be related to the mechanisms that underpin their transcriptional regulation. Transcription factors and developmental regulators are often subject to regulation by the polycomb repressive complex in early development (*Sawarkar and Paro, 2010*). Therefore the polycomb-mediated H3K27me3 chromatin modification was analysed at broad NMIs in mouse ES cells (*Mikkelsen et al., 2007*) and frog stage 11–12 embryos (*Akkers et al., 2009*) (*Figure 7E*). Strikingly 45% and 89% of broad NMIs were associated with H3K27me3 in mouse ES cells and frog stage 11–12 tissues respectively, a significantly higher proportion than observed for canonical NMIs (Fisher's exact test, odds ratio > 5.3, p<10⁻³⁵). As with H3K4me3, H3K27me3 extends across the broad region of non-methylated DNA suggesting that not only are broad NMIs preferentially subject to polycomb silencing but that polycomb complexes may also read the underlying non-methylated DNA state when placing H3K27me3 marks (*Figure 7F*). This idea is generally in agreement with a recent observation that clustered CGI predictions were often a good predictor of polycomb nucleation (*Orlando et al., 2012*). Although it still remains largely unknown how polycomb repressive complexes mechanistically recognise gene targets in vivo, our cross-species analysis reveals that polycomb repression is preferentially and spatially targeted to the broad class of NMIs. This reveals that there is a clear functional segregation between the canonical and broad class of NMI and provides an exciting new possibility that the capacity to function as a vertebrate polycomb responsive element may rely on properties specific to the broad class of NMI. Together this demonstrates that broad NMIs specify a unique subset of transcription factors and developmental regulators that are preferentially targeted for polycomb repression during early development, a process that appears to be highly conserved over vertebrate evolution.

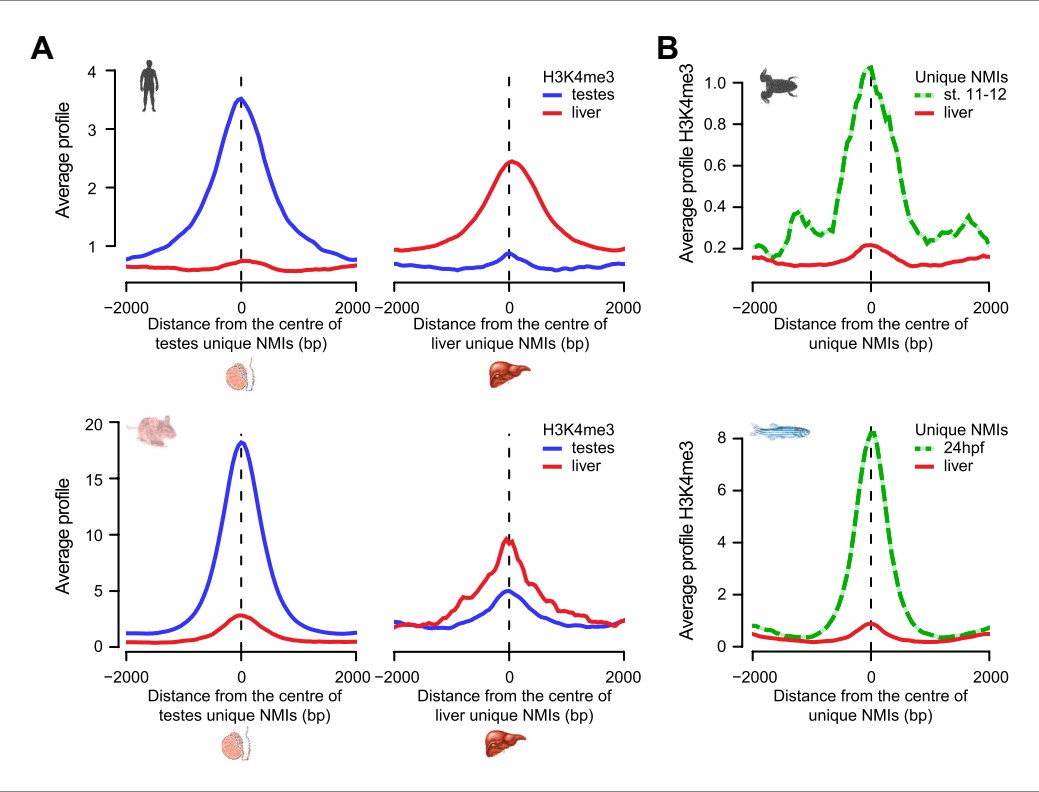

**Figure 6**. Chromatin modification at NMIs depends on their underlying DNA methylation state. (**A**) H3K4me3 read density from testes (blue) and liver (red) is profiled over testes unique (left) and liver unique (right) NMIs for human (upper) and mouse (lower) and displayed as an average profile. At differentially methylated loci, the histone H3K4me3 modification is found preferentially in the tissue with the non-methylated NMI. (**B**) The H3K4me3 signal (profiled in frog stage 11–12 embryos and zebrafish 24 hpf) is present specifically at unique NMIs from frog stage 11–12 and zebrafish 24 hpf (green) and not at unique NMIs from the liver (red).

## Discussion

Although the DNA methylation system is conserved across vertebrate evolution, CGI maps had previously indicated that this epigenetic system may have significantly diverged between vertebrate species and even acquired unique properties at TSSs during the evolution of warm-blooded verte-brates (*Aïssani and Bernardi, 1991*; *Sharif et al., 2010*). Despite some recent indications that DNA methylation profiles may be more conserved than previously realised (*Feng et al., 2010*; *Zemach et al., 2010*; *Wu et al., 2011*; *Andersen et al., 2012*), a lack of experimentally identified regions of non-methylated DNA outside of eutherian mammals has hindered the capacity to specifically address whether this system has significantly diverged among vertebrates. To address this fundamental ques-tion and to better understand the level of evolutionary conservation in epigenetic systems, we identi-fied NMIs genome-wide in seven diverse vertebrate species demonstrating for the first time that NMIs are in fact a highly conserved feature of vertebrate gene promoters. Importantly, this paradigm shift also revealed that three distinct yet highly conserved classes of NMIs have emerged during vertebrate evolution. The first class is a canonical NMI that best fits the classical definition of a CGI. These NMIs are narrow, associated with gene promoters, and generally remain free of DNA methylation regardless of the tissue or associated gene expression state. The second class of plastic NMIs are shorter, have lower CpG density than canonical NMIs, are usually found away from gene promoters at alternative regulatory elements, and are subject to differential methylation between tissues. Importantly, this class of NMI demonstrates that epigenetic plasticity in the form of differential methylation is a highly con-served mechanism utilised by all vertebrates. Finally, a third and unique class of broad NMIs were identified that often cover an entire gene, are specifically associated with transcription factors or developmental genes, and are associated with polycomb mediated silencing during early development.

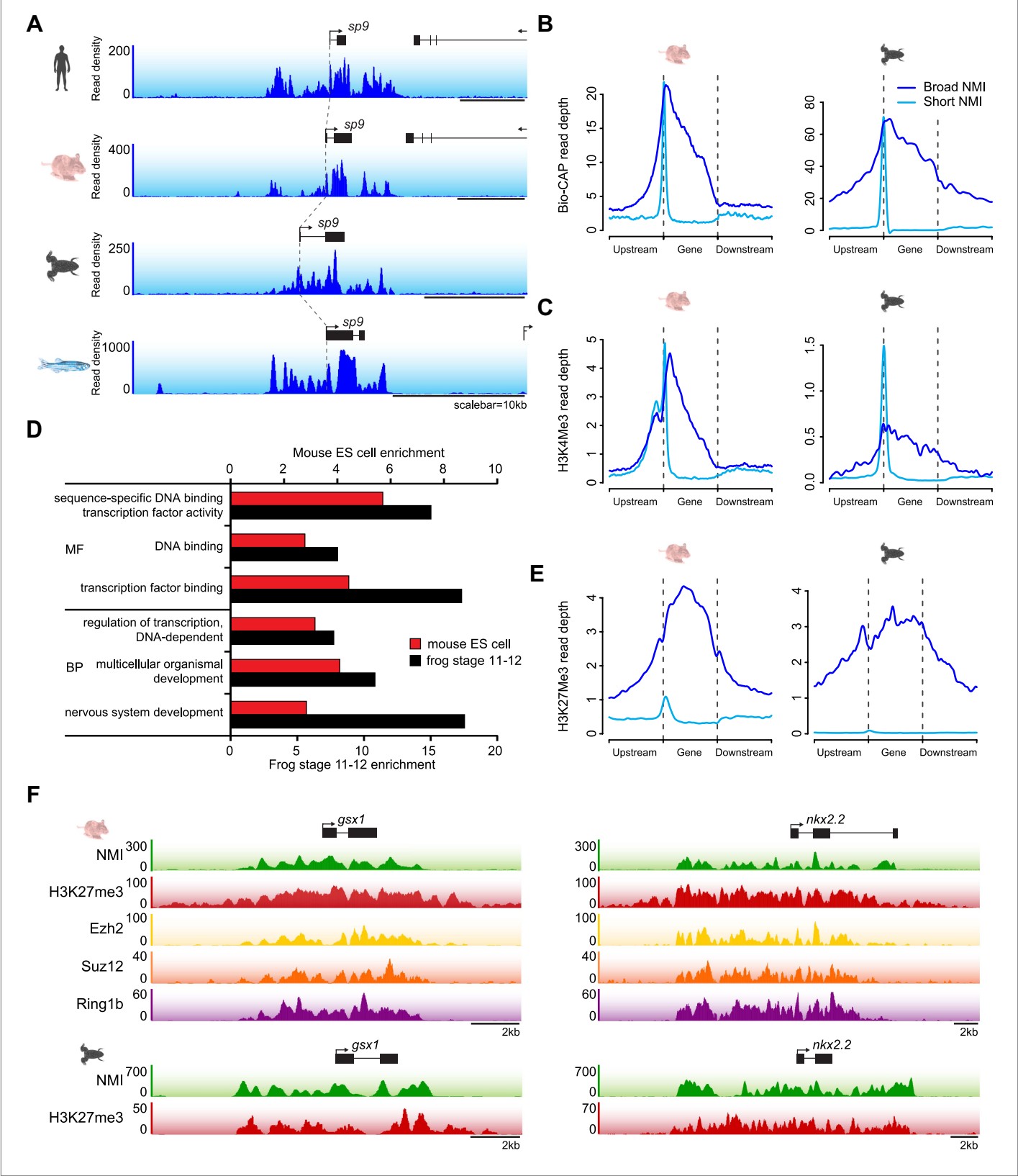

**Figure 7**. A unique class of broad non-methylated islands encompass polycomb-regulated developmental genes. (**A**) An example of a broad region of non-methylated DNA associated with the *sp9* gene for four representative species (human, mouse, frog and fish). Dashed grey lines highlight the location of the gene TSSs across the four species. (**B**) Non-methylated DNA profiles are depicted for genes associated with broad NMIs (dark blue) and
*Figure 7. Continued on next page*

*Figure 7. Continued*

canonical NMIs (light blue) in mouse embryonic stem (ES) cells and frog stage 11–12. The profile is scaled to show an averaged gene with one gene length depicted upstream and downstream. (**C**) H3K4me3 ChIP-seq signal from mouse and frog was plotted as in (**B**). H3K4me3 profiles reflect the underlying non-methylated DNA profiles. (**D**) Genes associated with broad NMIs were analysed by gene ontology (GO) analysis for mouse ES cell and frog stage 11–12. Broad NMIs are found to be significantly enriched for GO term categories associated with sequence-specific DNA binding, transcriptional regulation and development. MF: molecular function; BP: biological process. $p < 10^{-5}$ for all GO terms. (**E**) H3K27me3 ChIP-seq signal from mouse and frog was plotted for the same gene sets as in (**B**). The profile is scaled to show an averaged gene with three gene lengths depicted upstream and downstream. As for H3K4me3, H3K27me3 ChIP-seq profiles correspond to the underlying non-methylated DNA profile. (**F**) A representative example of two broadly non-methylated genes *gsx1* and *nkx2.2* for mouse and frog. In both species, the broad non-methylated regions (green) are associated with the polycomb repressive mark H3K27me3 (red). In addition, in mouse, polycomb repressive complex 2 (ezh2, yellow and suz12, orange) and polycomb repressive complex 1 (ring1b, purple) components are associated with the broad non-methylated regions. The y-axis depicts read density. Genes are depicted above the profiles in black.

The following figure supplements are available for figure 7:

**Figure supplement 1**. *Hox* gene clusters are characterized by broad NMIs.

These three classes of NMIs appear to form a highly conserved logic for the utilisation of non-methylated DNA in vertebrate genomes. Therefore, in contrast to the suggestion that NMIs may have diverged during vertebrate evolution (*Aïssani and Bernardi, 1991*; *Sharif et al., 2010*), we demonstrate that the central properties that underpin the NMI system are instead highly conserved across vertebrates. Perhaps most importantly, TSS-associated NMIs appears to be under strong selective pressure as part of what appears to be a highly conserved epigenetic system used to specify and control gene regulatory elements in large and complex vertebrate genomes.

## Materials and methods

### Preparation of genomic DNA

Samples were obtained either as purified genomic DNA, as fresh-frozen samples or were dissected in-house and fresh-frozen. Samples were subjected to manual homogenisation followed by DNA purification by phenol chloroform extraction or using the QIAGEN 100/G genomic tip kit (Manchester, UK).

### Bio-CAP sequencing

Bio-CAP was performed as previously described (*Blackledge et al., 2012*). All Bio-CAP experiments were performed in duplicate with matched input controls. Next generation sequencing was performed using two Illumina sequencing platforms: Genome Analyser IIx and HiSeq Systems yielding 51-bp single-end reads.

### External datasets

Computationally predicted CpG islands for all seven species were obtained from the UCSC genome browser (*Kent et al., 2002*). LncRNA datasets from recent publications for mouse (*Guttman et al., 2010*; *Belgard et al., 2011*; GSE20851, GSE27243) and zebrafish (*Ulitsky et al., 2011*; *Pauli et al., 2012*; GSE32900 and GSE32880) were obtained from GEO (*Edgar et al., 2002*) or from supplementary material. H3K4me1 datasets for zebrafish 24 hpf (*Aday et al., 2011*; GSE20600) and mouse ES cell (*Stadler et al., 2011*; GSE30206, GSE11172) were obtained from GEO. Mouse (*Smagulova et al., 2011*; GSE24438) and human (*Hammoud et al., 2009*; *Bernstein et al., 2010*; GSE15594 and GSE19465, testes and liver), frog (*Akkers et al., 2009*; GSE14025, stage 11–12) and zebrafish (*Aday et al., 2011*; GSE20600; 24 hpf) H3K4me3 datasets were obtained from GEO. H3K4me3 and H3K27Me3 datasets for mouse (*Mikkelsen et al., 2007*; GSE12241, ES cells) and frog (*Akkers et al., 2009*; GSE14025, stage 11–12) were obtained from GEO. RNAseq datasets for human, mouse, platypus and chicken (*Brawand et al., 2011*; *Stadler et al., 2011*; *Julien et al., 2012*; GSE30352, GSE30280 and GSE36120) were obtained from GEO. RNAseq datasets for lizard were obtained from Kutter and Odom pre-publication (*Barbosa-Morais et al., 2012*; now available at GSE41338) and zebrafish RNAseq data were obtained from the EBI (ERP000016, Sample ERS000081). The XTev dataset (*Akkers et al., 2010*) was obtained from the Veenstra lab website (http://131.174.221.43/gertjanveenstra/genomedata.asp). ChIP-seq data for a number of polycomb factors profiles for mouse ES cells (*Ku et al., 2008*; GSE13084) were obtained from GEO.

## Read alignment and peak calling

Sequencing reads were aligned to the appropriate reference genome (anoCar2, danRer7, galGal3, hg19, mm9, ornAna1, xenTro3) using the Bowtie short-read aligner (v0.12.7) (*Langmead et al., 2009*). Only uniquely mapping reads, with a maximum of two mismatches across the entire read length were used. Non-methylated islands (NMIs) were identified using MACS (v1.4.0) (*Zhang et al. 2008*) using a bandwidth of 300 and an mfold range of 10–30. Binding intervals were filtered by a q value of 0.01. Data analysis was performed in R and python using bespoke scripts available online (http://www.cgat.org/hg/cgat/).

## Nucleotide properties of NMIs

CpG observed/expected (CpG O/E) and GC content were calculated as in *Gardiner-Garden and Frommer (1987)*. Both measures were calculated for each NMI and for a control region of the same size 10 kb upstream of each NMI interval.

## NMI genomic localisation

Ensembl (release 66) genes were annotated as having an NMI at their TSS using a single base pair overlap of an NMI with a window extending 1 kb upstream and downstream from each transcript TSS. Multi-tissue RNAseq datasets for platypus, chicken and lizard were used to improve the annotation of gene TSSs. Where transcript models were not provided by the authors TopHat (*Trapnell et al., 2009*) and Cufflinks (*Trapnell et al., 2010*) were used to construct transcript models from short-read data. Similarly, the XTev gene dataset was used to improve the annotation of TSSs in the frog genome. NMIs were annotated with respect to both Ensembl (release 66) and RNAseq-based genome annotation as being associated with the following features in a hierarchical manner: protein-coding gene TSS (±1 kb), gene body, upstream or downstream of a gene (within 5 kb of the annotated gene model). Remaining NMIs were annotated as intergenic.

For mouse ES cell and zebrafish 24 hpf embryos, published lncRNA models, Ensembl non-coding RNA annotations, tissue-specific H3K4me1 and RNAseq data were used to account for intergenic NMIs in a hierarchical manner. Where short read genomic alignments were not provided by the authors, chromatin mark datasets were aligned to the appropriate reference genome using Bowtie and peaks were called using MACS as above. Throughout, overlap of genomic intervals (e.g., NMIs compared to CGIs, *Figure 1B*) was assessed using BEDTools (*Quinlan and Hall, 2010*) and statistical significance calculated using the Genomic Association Tester (GAT) (*Ponjavic et al., 2007*).

## Evolutionary conservation of NMIs

Evolutionary conservation of protein-coding genes was calculated using OPTIC (*Heger and Ponting, 2008*). Conserved genes (pairwise 1:1 orthologues) were defined as having an NMI or not as above. The conservation score was calculated as: n/min(x, y) where n is the number of conserved genes with an NMI at the TSS of both orthologues and x and y are the numbers of conserved genes with a TSS-associated NMI in each species. For three-way conservation, 1:1:1 orthologues from human, mouse and zebrafish were defined as having an NMI in one, two or all species.

## Tissue-specific & broad NMIs

NMIs were called from non-methylated DNA profiles in both testes and liver using MACS (as above). An NMI was defined as tissue-specific if it did not overlap with an NMI in the other tissue. Broad NMI-associated genes were defined as having greater than 90% of their gene length covered by NMIs. Short NMI-associated genes had less than 10% (but greater than 0%) gene coverage by NMIs.

## Gene expression analysis

DESeq (*Anders and Huber, 2010*) was used to identify genes differentially expressed between liver and testes in human, mouse, platypus and chicken RNAseq data (p<0.05, fold change > 2).

## Data visualisation

H3K4me3 signal was profiled across tissue-specific NMIs using sitepro from the CEAS package (*Shin et al., 2009*). Two-way Venn diagrams were generated in R using the 'VennDiagram' package (*Chen and Boutros, 2011*). The three-way Venn diagram was generated using the EulerAPE drawing tool

(http://www.eulerdiagrams.org/eulerAPE/). Genomic peaks and intervals were visualised using the Integrated Genome Browser (IGB) (*Nicol et al., 2009*).

## Gene ontology

Gene Ontology (GO) analysis was performed using a hypergeometic test. Terms were clustered using a modified ReVigo (*Supek et al., 2011*) script and a representative term from each cluster was plotted using the GO term enrichment score.

## Acknowledgements

We thank the High-Throughput Genomics Group at the Wellcome Trust Centre for Human Genetics (funded by Wellcome Trust grant reference 090532/Z/09/Z and MRC Hub grant G0900747 91070) for the generation of the sequencing data, especially Lorna Gregory for her technical support. For samples, we would like to thank Jonathan Godwin for mouse liver and testes tissue, Mike Colley from FAI Farms for chicken liver and testes tissue, Jonathan Losos and Shane Campbell-Staton for lizard liver DNA and testes tissue, Anita Abu-Daya for help collecting stage 11–12 frog embryos and Anna Noble for frog testes tissue. We would also like to thank Stephen Watt and Nuno Barbosa-Morais for lizard RNA-seq data analysis, and Anamaria Necsulea for providing platypus and chicken gene models from RNAseq data. We are also grateful to Anca Farcas and Nathan Rose for critical reading of the manuscript and to Neil Brockdorff and his laboratory for discussion. Datasets generated in this study are available from GEO under accession GSE43512.

## Additional information

### Competing interests

CPP: Senior Editor, *eLife*. The other authors declare that no competing interests exist.

### Funding

| Funder | Grant reference number | Author |
|---|---|---|
| Wellcome Trust | WT0834922 | Robert J Klose, Neil P Blackledge |
| Wellcome Trust | 090018/Z/09/Z | Hannah K Long |
| Cancer Research UK | | Claudia Kutter, Duncan T Odom |
| Medical Research Council | G1000902 | Chris P Ponting, David Sims, Andreas Heger |
| Lister Institute of Preventive Medicine | | Robert J Klose |
| EMBO | | Robert J Klose, Duncan T Odom |
| European Research Council Starting Grant | 202218 | Duncan T Odom |
| Medical Research Council | | Roger Patient |
| Swiss National Science Foundation | | Claudia Kutter |
| ARC Australian Research Fellowship | | Frank Grützner |
| Australian Postgraduate Award | | Megan L Wright |

The funders had no role in study design, data collection and interpretation, or the decision to submit the work for publication.

### Author contributions

HKL, Conception and design, Acquisition of data, Analysis and interpretation of data, Drafting or revising the article; DS, AH, CPP, RJK, Conception and design, Analysis and interpretation of data, Drafting or revising the article; NPB, RP, Drafting or revising the article; CK, MLW, FG, DTO, Drafting or revising the article, Contributed unpublished essential data or reagents

# Additional files

## Major datasets

The following dataset was generated:

| Author(s) | Year | Dataset title | Dataset ID and/or URL | Database, license, and accessibility information |
|---|---|---|---|---|
| Long HK, Sims D, Heger A, Blackledge NP, Kutter C, Wright ML, Grützner F, Odom DT, Patient R, Ponting CP, Klose RJ | 2013 | Epigenetic conservation at gene regulatory elements revealed by non-methylated DNA profiling in seven vertebrates | GSE43512; http://www.ncbi.nlm.nih.gov/geo/query/acc.cgi?acc=GSE43512 | In the public domain at GEO: http://www.ncbi.nlm.nih.gov/geo/. |

The following previously published datasets were used:

| Author(s) | Year | Dataset title | Dataset ID and/or URL | Database, license, and accessibility information |
|---|---|---|---|---|
| Mikkelsen TS, Meissner A, Zhang X, Gnirke A, Jaenisch R, Lander ES | 2007 | Genome-wide chromatin state maps of ES cells, ES-derived neural progenitor cells and brain tissue | GSE11172; http://www.ncbi.nlm.nih.gov/geo/query/acc.cgi?acc=GSE11172 | In the public domain at GEO: http://www.ncbi.nlm.nih.gov/geo/. |
| Mikkelsen TS, Ku M, Koche RP, Rheinbay E, Cowan CA, Lander ES, Bernstein BE | 2008 | Mapping polycomb complexes in human and mouse embryonic stem cells | GSE13084; http://www.ncbi.nlm.nih.gov/geo/query/acc.cgi?acc=GSE13084 | In the public domain at GEO: http://www.ncbi.nlm.nih.gov/geo/. |
| Akkers RC, van Heeringen SJ, Jacobi UG, Janssen-Megens EM, Françoijs K, Stunnenberg HG, Veenstra GJ | 2009 | A Hierarchy of H3K4me3 and H3K27me3 Acquisition in Spatial Gene Regulation in Xenopus Embryos | GSE14025; http://www.ncbi.nlm.nih.gov/geo/query/acc.cgi?acc=GSE14025 | In the public domain at GEO: http://www.ncbi.nlm.nih.gov/geo/. |
| Hammoud SS, Nix DA, Zhang H, Purwar J, Carrell DT, Cairns BR | 2009 | Distinctive Chromatin in Human Sperm Packages Genes that Guide Embryo Development | GSE15594; http://www.ncbi.nlm.nih.gov/geo/query/acc.cgi?acc=GSE15594 | In the public domain at GEO: http://www.ncbi.nlm.nih.gov/geo/. |
| Bernstein BE, Meissner A | 2010 | BI Human Reference Epigenome Mapping Project: ChIP-Seq in human subject | GSE19465; http://www.ncbi.nlm.nih.gov/geo/query/acc.cgi?acc=GSE19465 | In the public domain at GEO: http://www.ncbi.nlm.nih.gov/geo/. |
| Guttman M, Garber M, Lander E, Regev A | 2010 | Ab initio reconstruction of transcriptomes of pluripotent and lineage committed cells reveals gene structures of thousands of lincRNAs | GSE20851; http://www.ncbi.nlm.nih.gov/geo/query/acc.cgi?acc=GSE20851 | In the public domain at GEO: http://www.ncbi.nlm.nih.gov/geo/. |
| Aday AW, Zhu LJ, Lawson ND, Lakshmanan A | 2011 | Genome-wide identification of putative cis-regulatory elements through epigenetic profiling in zebrafish | GSE20600; http://www.ncbi.nlm.nih.gov/geo/query/acc.cgi?acc=GSE20600 | In the public domain at GEO: http://www.ncbi.nlm.nih.gov/geo/. |
| Belgard TG, Marques AC, Oliver PL, Ozel Abaan H, Sirey TM, Garcia-Moreno F, Molnar Z, Margulies EH, Ponting CP | 2011 | A Transcriptomic Atlas of Mouse Neocortical Layers | GSE27243; http://www.ncbi.nlm.nih.gov/geo/query/acc.cgi?acc=GSE27243 | In the public domain at GEO: http://www.ncbi.nlm.nih.gov/geo/. |
| Brawand D, Soumillon M, Necsulea A, Julien P, Csárdi G, Harrigan P, Weier M, Liechti A, Aximu-Petri A, Kircher M, Albert FW, Zeller U, Khaitovich P, Grützner F, Bergmann S, Nielsen R, Pääbo S, Kaessmann H | 2011 | The evolution of gene expression levels in mammalian organs | GSE30352; http://www.ncbi.nlm.nih.gov/geo/query/acc.cgi?acc=GSE30352 | In the public domain at GEO: http://www.ncbi.nlm.nih.gov/geo/. |
| Smagulova F, Gregoretti IV, Brick KM, Khil P, Camerini-Otero RD, Petukhova GV | 2011 | Genome wide maps of Dmc1 in testis of Hop2 null mice | GSE24438; http://www.ncbi.nlm.nih.gov/geo/query/acc.cgi?acc=GSE24438 | In the public domain at GEO: http://www.ncbi.nlm.nih.gov/geo/. |

| | | | | |
|---|---|---|---|---|
| Stadler MB, Murr R, Burger L, Ivanek R, Lienert F, Anne S, Oakeley EJ, Gaidatzis D, Tiwari V, Schubeler D | 2011 | DNA binding factors shape the mouse methylome at distal regulatory regions [RNA_seq] | GSE30280; http://www.ncbi.nlm.nih.gov/geo/query/acc.cgi?acc=GSE30280 | In the public domain at GEO: http://www.ncbi.nlm.nih.gov/geo/. |
| Stadler MB, Murr R, Burger L, Ivanek R, Lienert F, Scholer A, Oakeley EJ, Gaidatzis D, Tiwari V, Schubeler D | 2011 | DNA binding factors shape the mouse methylome at distal regulatory regions | GSE30206; http://www.ncbi.nlm.nih.gov/geo/query/acc.cgi?acc=GSE30206 | In the public domain at GEO: http://www.ncbi.nlm.nih.gov/geo/. |
| Ulitsky I, Shkumatava A, Jan CH, Sive H, Bartel DP | 2011 | Conserved Function of lincRNAs in Vertebrate Embryonic Development Despite Rapid Sequence Evolution | GSE32880; http://www.ncbi.nlm.nih.gov/geo/query/acc.cgi?acc=GSE32880 | In the public domain at GEO: http://www.ncbi.nlm.nih.gov/geo/. |
| Julien P, Brawand D, Soumillon M, Necsulea A, Liechti A, Daish T, Grützner F, Kaessmann H | 2012 | Platypus fibroblast and ovary transcriptomes | GSE36120; http://www.ncbi.nlm.nih.gov/geo/query/acc.cgi?acc=GSE36120 | In the public domain at GEO: http://www.ncbi.nlm.nih.gov/geo/. |
| Pauli A, Valen E, Lin MF, Garber M, Vastenhouw NL, Levin JZ, Sandelin A, Rinn JL, Regev A, Schier AF | 2012 | Comprehensive identification of long non-coding RNAs expressed during zebrafish embryogenesis | GSE32900; http://www.ncbi.nlm.nih.gov/geo/query/acc.cgi?acc=GSE32900 | In the public domain at GEO: http://www.ncbi.nlm.nih.gov/geo/. |
| The Wellcome Trust Sanger Institute | | Sequencing the Zebrafish transcriptome form a range of tissues and developmental stages using the less Illumina Genome Analyzer | ERP000016, Sample ERS000081; http://www.ebi.ac.uk/ena/data/view/ERS000081 | Available at http://www.ebi.ac.uk/ena/data/view/ERP000016&display=html. |
| Barbosa-Morais NL, Kutter C, Watt S, Odom DT, Blencowe BJ | 2012 | The evolutionary landscape of alternative splicing in vertebrate species | GSE41338; http://www.ncbi.nlm.nih.gov/geo/query/acc.cgi?acc=GSE41338 | In the public domain at GEO: http://www.ncbi.nlm.nih.gov/geo/. |

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
