## [Decision Letter]

Thank you for choosing to send your work entitled “Epigenetic conservation at gene regulatory elements revealed by non-methylated DNA profiling in seven vertebrates” for consideration at *eLife*. Your article has been evaluated by a Senior editor and 3 reviewers, one of whom is a member of our Board of Reviewing Editors. The following individuals responsible for the peer review of your submission want to reveal their identity: Anne Ferguson-Smith (Reviewing editor); Wolf Reik (peer reviewer).

The Reviewing editor and the other reviewers discussed the manuscript before we reached this decision, and the Reviewing editor has assembled the following comments based on the individual reports.

General assessment: this is a well executed, clearly presented original piece of work and publication is recommended.

Comments to address:

1. Please validate some of the new NMIs by orthogonal, quantitative single locus studies, such as bisulphite sequencing, pyrosequencing or EpiTyping to support the idea that the method used by the authors confidently identifies NMIs. This is mentioned as data not shown but please provide some examples.

2. Could the authors briefly summarise (perhaps in the Introduction) why the different species have such different sequence composition, leading to the problem with the prediction algorithm. Also, the performance of the affinity-based assay may differ between species depending on the CpG content overall and at CpG-dense loci. This should be acknowledged.

3. The substantial variability between species of genomic CpG and (G+C) content has previously been described and should be cited (for example PMID: 17932072), as it implies the need for species-specific CpG island annotation rather than the UCSC approach.

4. It is not clear why the authors talk about NMIs at promoters being “contrary to expectations”, when the opposite is probably more generally accepted. Please justify or change this.

5. Although this does not influence the conclusions, it should be acknowledged that an affinity approach (such as the use of the CXXC reagent) is more likely to be sensitive to CpG-dense than CpG-depleted loci.

6. Regarding the UCSC CpG island annotation, it should be recognized that the annotation is performed on repeat masked sequence, so the quality of the annotation of repetitive elements also plays a role in CpG island annotation (in addition to reference sequence quality).

---

## [Author Response]

*1. Please validate some of the new NMIs by orthogonal, quantitative single locus studies, such as bisulphite sequencing, pyrosequencing or EpiTyping to support the idea that the method used by the authors confidently identifies NMIs. This is mentioned as data not shown but please provide some examples*.

As requested by the reviewers we have now included bisulphite sequencing analysis to support the idea that Bio-CAP confidently identifies NMIs. This is included in the new figure, Figure 5–figure supplement 1', and we now refer to this figure in the manuscript in place of ‘data not shown’. In all cases the bisulfite sequencing demonstrates that NMIs identified by Bio-CAP signal correspond to regions containing non-methylated DNA. This is in agreement with our previous work describing the Bio-CAP technique (Blackledge NP and Long HK, NAR, 2012). Furthermore, we demonstrate using bisulphite sequencing that unique NMIs between testes and liver in both mouse and zebrafish correspond to regions with differential methylation.

*2. Could the authors briefly summarise (perhaps in the Introduction) why the different species have such different sequence composition, leading to the problem with the prediction algorithm. Also, the performance of the affinity-based assay may differ between species depending on the CpG content overall and at CpG-dense loci. This should be acknowledged*.

We agree with the reviewers that it would be useful to discuss in more detail why different species have different sequence composition. However, we thought it would be more appropriate to cover this in the section entitled ‘Nucleotide properties within NMIs are variable in different vertebrate genomes’ than in the Introduction. We have added text to explain how species acquire different sequence composition and also added a statement qualifying that Bio-CAP is an affinity based approach and therefore this could affect performance between species.

*3. The substantial variability between species of genomic CpG and (G+C) content has previously been described and should be cited (for example PMID: 17932072), as it implies the need for species-specific CpG island annotation rather than the UCSC approach*.

We agree with the reviewer that the variability in nucleotide content amongst species has previously been identified as a hurdle to accurate bioinformatic prediction of non-methylated islands using the UCSC based approach. We have added text to the ‘Nucleotide properties within NMIs are variable in different vertebrate genomes’ section of the manuscript to clarify this point and we have cited the suggested reference.

*4. It is not clear why the authors talk about NMIs at promoters being “contrary to expectations”, when the opposite is probably more generally accepted. Please justify or change this*.

In warm-blooded vertebrates, including human and mouse, more than half of all gene transcription start sites (TSSs) are associated with CGIs. In contrast, CGI predictions in cold-blooded vertebrates, including frog and zebrafish, are infrequently associated with gene TSSs. This potentially represents a major difference in promoter architecture between warm-blooded and cold-blooded vertebrates, and has been used previously as an argument that CGIs may have arisen as a feature in the gene promoters of endotherms. We use a phrase similar to “contrary to expectation” a number of times during the manuscript to refer to the fact that CGI predictions, and their usage in comparative evolutionary studies, suggest that CGI usage, and in warm- and cold-blooded vertebrates, may be divergent.

In retrospect we see that it would be more useful to clarify this point early in the manuscript, and we have now added sections to the revised manuscript that illustrates this point quantitatively and clarifies the differences in CGI prediction between warm- and cold-blooded vertebrates within the Introduction and within the section ‘NMIs are a highly conserved feature of vertebrate gene promoters’ section.

As indicated by the reviewers this concept is iterated in five places throughout the manuscript. We have now clarified this point or further supported it by references in all instances.

*5. Although this does not influence the conclusions, it should be acknowledged that an affinity approach (such as the use of the CXXC reagent) is more likely to be sensitive to CpG-dense than CpG-depleted loci*.

We have included the following text to emphasise this point:

“Bio-CAP identifies NMIs through an affinity based isolation of non-methylated CpGs and therefore does not solely rely on nucleotide content in the same way prediction algorithms do. Nevertheless, it does remain possible that the efficiency of NMI identification by Bio-CAP between species may differ due to the overall non-methylated CpG content and density. However this does not appear to be the case as non-methylated DNA fragments, even with low CpG density, are effectively detected by Bio-CAP (Blackledge et al, 2012) and a broad distribution of CpG density within NMIs is identified in all species.”

*6. Regarding the UCSC CpG island annotation, it should be recognized that the annotation is performed on repeat masked sequence, so the quality of the annotation of repetitive elements also plays a role in CpG island annotation (in addition to reference sequence quality)*.

We have included a statement in the ‘Nucleotide properties within NMIs are variable in different vertebrate genomes’ section to emphasise that the quality of repeat element annotation will also affect the accuracy of CGI prediction:

“The failure of CpG island prediction algorithms to accurately identify NMIs in different species is almost certainly dependent on the variation in CpG density and G+C content amongst vertebrate genomes, but also will rely on genome assembly and annotation quality, particularly of repetitive elements.”